# Correlation between Hepatitis B Virus Infection and Colorectal Neoplasia

**DOI:** 10.3390/jcm8122085

**Published:** 2019-12-01

**Authors:** Yoon Suk Jung, Nam Hee Kim, Jung Ho Park, Dong Il Park, Chong Il Sohn

**Affiliations:** 1Division of Gastroenterology, Department of Internal Medicine, Kangbuk Samsung Hospital, Sungkyunkwan University School of Medicine, Seoul 03181, Korea; jungho3.park@samsung.com (J.H.P.); diksmc.park@samsung.com (D.I.P.); chongil.sohn@samsung.com (C.I.S.); 2Preventive Healthcare Center, Kangbuk Samsung Hospital, Sungkyunkwan University School of Medicine, Seoul 03181, Korea; knh83love@naver.com

**Keywords:** Colorectal neoplasia, Hepatitis B, Hepatitis C

## Abstract

Background: Data about the association between hepatitis virus infection and colorectal neoplasia (CRN) are extremely limited. We examined the association between hepatitis B virus (HBV) and hepatitis C virus (HCV) infection with the risk of CRN. Methods: A cross-sectional study was performed on asymptomatic examinees who underwent a colonoscopy and serologic testing for hepatitis B surface antigen (HBsAg) and hepatitis C antibody (HCV Ab) between 2004 and 2015. Results: Of 155,674 participants who underwent serologic testing for HBsAg, 5476 (3.5%) were positive for HBsAg. The mean age of the study participants was 41.1 ± 9.1 years. The prevalence of CRN was higher in the HBsAg (+) than in HBsAg (-) participants (16.9% vs. 15.6%, *p* = 0.009). Even after adjusting for confounders, HBsAg positivity was correlated with an increased risk of CRN (odds ratio (OR), 1.10; 95% confidence interval (CI), 1.01–1.19; *p* = 0.025). Of 155,180 participants who underwent serologic testing for HCV Ab, only 240 (0.15%) were positive for HCV Ab. The prevalence of CRN was higher in HCV Ab (+) than in HCV Ab (-) participants (22.9% vs. 15.6%, *p* = 0.002). However, the association disappeared after adjusting for confounders (OR, 1.04; 95% CI, 0.72–1.50; *p* = 0.839). Conclusions: HBV infection was independently correlated with an increased risk of CRN. Our results indicate the possibility that HBV infection may contribute to colorectal carcinogenesis. Screening colonoscopy may have to be recommended more thoroughly for HBV-infected patients.

## 1. Introduction

Chronic viral hepatic infections are a major threat to worldwide public health [1]. Hepatitis B virus (HBV) and hepatitis C virus (HCV) are the major causes of chronic hepatitis, and they are known to be involved in the etiologies of cirrhosis and hepatocellular carcinoma (HCC) [1]. In 2015, the global prevalence of HBV infection in the general population was estimated at 3.5%, with about 257 million persons living with chronic HBV infection [2]. HCV has been estimated to chronically infect approximately 160 million people worldwide, with a global prevalence of 2.4% [3].

Although HBV is considered a hepatotropic virus, several studies have reported that HBV can exist in extrahepatic organs including the pancreas, lung, kidney, skin, lymph nodes, spleen, and bone marrow [4,5]. Similarly, several other studies have shown that pancreatic, lung, kidney, and skin cancers; lymphoma; and leukemia are linked with chronic HBV infection [6,7,8,9]. Additionally, HBV has been reported to exist in the gastrointestinal tract, including the stomach and colon [4,10]. A recent study demonstrated that chronic HBV infection is associated with the risk of gastric cancer [11], and another study revealed that chronic HBV infection is associated with the risk of colorectal cancer (CRC) [12]. However, studies on the association between HBV infection and colorectal adenoma, a precancerous lesion of CRC, are lacking. Thus far, two studies have investigated this topic; however, the sample size was too small to make a definitive conclusion [13,14].

Meanwhile, some studies have revealed a link between HCV infection and extrahepatic cancers such as immunoproliferative malignancies [9,15]. However, little is known about the effects that HCV infection can have on the carcinogenesis of other common malignancies such as CRC. Although two previous studies have examined the association of HCV infection with the risk of colorectal adenoma, the results were inconsistent [16,17]. Drawing a reliable conclusion from the results of these studies is difficult.

Therefore, the aim of the present study was to evaluate the association between HBV infection (hepatitis B surface antigen (HBsAg) positivity) or HCV infection (hepatitis C antibody (HCV Ab) positivity) and the risk of colorectal neoplasia (CRN), including colorectal adenoma and CRC, in a very large sample of asymptomatic examinees (*n* = 197,050). We sought to identify whether HBsAg positivity or HCV Ab positivity is independently associated with the risk of CRN.

## 2. Methods

### 2.1. Study Population

The Kangbuk Samsung Health Study is a cohort study of South Korean men and women aged 18 years or older, who undergo a comprehensive annual or biennial health examination at the clinics of the Kangbuk Samsung Hospital Total Healthcare Center in Seoul and Suwon, South Korea. The study population consisted of a subset of Kangbuk Samsung Health Study participants who underwent a colonoscopy and serologic testing for HBsAg or HCV Ab as part of a comprehensive health examination from August 2004 to December 2015. In South Korea, the Industrial Safety and Health Law requires annual or biennial health screening examinations of all employees, which are offered free of charge. Approximately, 80% of the participants were employees of various companies and local governmental organizations or their spouses, and the remaining participants were those who volunteered to participate in the screening examinations.

The exclusion criteria were as follows: a history of CRC or colorectal surgery, a history of colon examination or colon polyp removal, a history of inflammatory bowel disease, a history of HCC, and poor bowel preparation. Poor bowel preparation was defined as large amounts of solid fecal matter found, precluding a satisfactory study; unacceptable preparation; or < 90% of the mucosa seen [18].

This study was approved by the Institutional Review Board of Kangbuk Samsung Hospital (KBSMC 2017-03-025). The requirement for informed consent was waived because only de-identified data were retrospectively assessed.

### 2.2. Measurements and Definitions

Data on medical history and health-related behaviors was collected through a self-administered questionnaire, and physical measurements were obtained by trained staff. Smoking status was categorized as never, former, and current, and alcohol intake was estimated in grams per day. Family history of CRC was defined as CRC in ≥ 1 first-degree relative at any age. Self-reported use of nonsteroidal anti-inflammatory drugs (NSAIDs) regularly over the past month was assessed. Obesity was defined as a body mass index of ≥30 kg/m^2^.

The presence of HBsAg and HCV Ab was determined using immunoradiometric assays (Radim, Via del Mare, Italy) in the Seoul center from 2004 to 2009 and in the Suwon center from 2004 to 2006, and using electrochemiluminescent immunoassays (Modular E170; Roche Diagnostics, Tokyo, Japan) since 2010 in the Seoul center and since 2007 in the Suwon center.

The presence of fatty liver was examined using abdominal ultrasonography with a 3.5-MHz transducer (LOGIQ 9; General Electric, Madison, WI, USA), performed by experienced radiologists who were unaware of the study aims. An ultrasonographic diagnosis of fatty liver was defined as the presence of a diffuse increase in fine echoes in the liver parenchyma compared with the kidney or spleen parenchyma [19]. The interobserver and intraobserver reliability values for the diagnosis of fatty liver were very high (kappa statistics of 0.74 and 0.94, respectively) [20].

### 2.3. Colonoscopy and Histologic Examination

All colonoscopic examinations were performed using the EVIS LUCERA CV-260 colonoscope (Olympus Medical Systems, Tokyo, Japan) by board-certified endoscopists. Bowel cleansing was performed using 4 L of polyethylene glycol solution, and a split-dose preparation was employed. Suspicious neoplastic lesions were examined by taking a biopsy sample, or were removed through polypectomy or endoscopic mucosal resection. All specimens were histopathologically assessed by experienced gastrointestinal pathologists. Overall CRN was defined as cancer or adenoma. Advanced adenoma was defined as the presence of one of the following features: ≥10 mm diameter, tubulovillous or villous structure, and high-grade dysplasia [21].

### 2.4. Statistical Analysis

The demographic characteristics and the prevalence of CRN were compared according to the HBsAg and HCV Ab status. Data are expressed as mean ± standard deviation or frequency (%). Student’s t tests were used to compare continuous variables between the groups, and the chi-square tests or Fisher exact tests were used to compare categorical variables between the groups. We estimated the odds ratios (ORs) and 95% confidence intervals (CIs) for CRN prevalence according to the HBsAg or HCV Ab status, through logistic regression. Multivariable models were adjusted for the following potential confounders: age, sex smoking status, family history of CRC, obesity, alcohol intake, fatty liver, hypertension, diabetes, and use of NSAIDs. All reported *p*-values were 2-tailed, and *p* < 0.05 was considered statistically significant. The software program SPSS version 18 (SPSS Inc., Chicago, IL, USA) was used for statistical analyses.

## 3. Results

### 3.1. Baseline Characteristics of the Study Population

We reviewed the medical records of 197,050 participants who underwent colonoscopy and serologic testing for HBsAg or HCV Ab. Of them, 41,376 were excluded because of a history of colorectal surgery or CRC (*n* = 558), a history of colon examination or colon polyp removal (*n* = 22,709), a history of inflammatory bowel disease (*n* = 357), a history of HCC (*n* = 44), and poor bowel preparation (*n* = 17,708). Ultimately, 155,674 participants were analyzed. The mean age of the study participants was 41.1 ± 9.1 years, and the proportion of men was 65.3%.

Of the 155,674 participants, 5476 (3.5%) were positive for HBsAg. Table 1 shows the demographic characteristics of HBsAg (+) and HBsAg (-) participants. Compared with HBsAg (-) participants, HBsAg (+) participants were more likely to be older, male, and smokers; to have a higher BMI; to consume less alcohol; and to use NSAIDs less. The prevalence of fatty liver was higher in seronegative participants than in seropositive participants. The prevalence of overall CRN was higher in HBsAg (+) participants than in HBsAg (-) participants (16.9% vs. 15.6%, *p* = 0.009). However, the prevalence of advanced adenoma and CRC were not different between HBsAg (-) and HBsAg (+) participants (advanced adenoma: 1.8% vs. 1.9%, *p* = 0. 503 and CRC: 0.07% vs. 0.07%, *p* = 1.000). 

Of the 155,674 participants, 155,180 underwent serologic testing for HCV Ab and 494 did not. Of the 155,180 participants who underwent serologic testing for HCV Ab, 240 (0.15%) were positive for HCV Ab. Eight participants were positive for both HBsAg and HCV. The demographic characteristics of HCV Ab (+) and HCV Ab (-) participants are compared in Table 2. Compared with HCV Ab (-) participants, HCV Ab (+) participants were more likely to be older and female. The prevalence of fatty liver was higher in HCV Ab (-) participants than in HCV Ab (+) participants, whereas the prevalence of diabetes and hypertension were higher in seropositive participants than in seronegative participants. The prevalence of overall CRN was higher in HCV Ab (+) participants than in HCV Ab (-) participants (22.9% vs. 15.6%, *p* = 0.002). However, there were no significant differences in the prevalence of advanced adenoma and CRC between the 2 groups.

### 3.2. Association between Hepatitis B or C Virus Infection and Colorectal Neoplasia

Table 3 shows the risk of overall CRN due to hepatitis virus infection. The crude (unadjusted) OR for prevalent CRN comparing HBsAg (+) to HBsAg (-) participants was 1.10 (95% CI, 1.02–1.18; *p* = 0.009) and that for comparing HCV Ab (+) to HCV Ab (-) participants was 1.30 (95% CI, 1.19–2.17; *p* = 0.002). Even after adjusting for major confounding factors, the association between CRN risk and HBV infection remained significant (OR, 1.10; 95% CI, 1.01–1.19; *p* = 0.025). However, the association between CRN risk and HCV infection disappeared after adjusting for confounders (OR, 1.04; 95% CI, 0.72–1.50; *p* = 0.839).

We examined the risk factors for CRN among HBsAg (+) participants (Table 4). In multivariate analysis adjusted for confounders, age (OR, 1.08; 95% CI, 1.07–1.09), male sex, (OR, 1.51; 95% CI, 1.18–1.95), current or former smoking (OR, 1.57; 95% CI, 1.29–1.92), family history of CRC (OR, 1.94; 95% CI, 1.33–2.84), alcohol intake (OR, 1.23; 95% CI, 1.02–1.49), and fatty liver (OR, 1.24; 95% CI, 1.04–1.48) were identified as risk factors for CRN in HBsAg (+) participants.

## 4. Discussion

In this large study of healthy Korean men and women, we found a cross-sectional correlation of HBV infection with the prevalence of CRN. The correlation was evident even after adjusting for confounding factors. However, we did not find an association between HCV infection and the risk of CRN. Although HCV Ab positivity was associated with a higher risk of CRN in univariate analysis, the association did not remain significant in multivariate analysis. Our study adds clear evidence regarding the correlation between HBV infection and the risk of CRN, and suggests that HBV infection may be involved in the etiology of CRN. On the basis of our results, HBV-infected patients may benefit from more intensive CRC screening. For example, screening colonoscopy rather than fecal immunochemical tests may have to be recommended for HBV-infected patients.

To date, only a few studies have investigated the association between HBV infection and CRN risk. Two previous studies showed an association between HBV and colorectal adenoma. Patel et al. reported that although the adenoma detection rate was higher in the hepatitis B group than in the non-hepatitis B group, the difference did not reach statistical significance (23.9%, *n* = 17/71 vs. 16.4%, *n* = 80/487; OR, 1.33; 95% CI, 0.70–2.51; *p* = 0.38) [13]. However, they demonstrated that there was a statistically significant association between HBV infection and the presence of distal colorectal adenomas (OR, 2.16; 95% CI, 1.06–4.43; *p* = 0.04) [13]. Recently, Kim et al. showed that HBV infection (23.3%, *n* = 31/133 vs. 1.5%, *n* = 6/399; adjusted OR, 24.0; 95% CI, 9.40–61.1; *p* < 0.001) and HBV DNA in patients with HBV infection (HBV DNA (log); adjusted OR, 1.24; 95% CI, 1.03–1.49; *p* = 0.024) were independently associated with the risk of advanced adenoma [14]. However, the prior 2 studies are limited by their small sample size (*n* = 558 and *n* = 532, respectively). Our study including a relatively large sample size (*n* = 155,674) provides reliable evidence on an association between HBV infection and CRN.

The literature on the relationship between HCV and CRN is also limited. Thus far, there are only two studies on this topic and the findings were inconsistent. Similar to our results, one study revealed that there was no statistical difference in the prevalence of colorectal adenoma between the HCV group and the control group (26.3%, *n* = 46/175 vs. 20.2%, *n* = 204/1008; adjusted OR, 1.02; 95% CI, 0.66–1.57; *p* = 0.93) [16]. However, unlike our findings, those of Rustagi et al. showed that HCV infection was an independent risk factor for colorectal adenoma (37.8%, *n* = 88/233 vs. 30.3%, *n* = 141/466; adjusted OR, 1.47; 95% CI, 1.01–2.14), particularly advanced neoplasia (14.6%, *n* = 34/233 vs. 9.2%, *n* = 43/466; adjusted OR, 2.04; 95% CI, 1.20–3.49) [17]. Rustagi et al. suggested the existence of a link between HCV infection and a higher risk of colorectal adenoma occurring through a T-cell-mediated process [17]. In addition, they proposed that HCV core protein may function as a gene regulator, with the ability to inhibit the tumor suppressor gene *p53* and induce the growth factor nuclear factor-κB [22]. The reason for the different results between the studies on HCV may be the small number of patients with HCV. The number of participants with HCV Ab in our study was also relatively small (*n* = 240, 0.15%). The prevalence of HCV Ab positivity in the general Korean population ranges from 0.5% to 1.0%, with the highest prevalence in those > 60 years old [23,24]. The prevalence of HCV Ab (+) in Korea is lower than the global standards (global estimates of prevalence, 2.35%) [3]. In our study, the prevalence of CRN was higher in HCV Ab (+) participants than in HCV Ab (-) participants (22.9% vs. 15.6%, *p* = 0.002). However, these results could be because HCV Ab (+) participants were much older than HCV Ab (-) participants (mean age, 50 vs. 41 years). Age is a potent risk factor for CRN [25]. Additionally, HCV Ab (+) participants had a higher prevalence of diabetes and hypertension, which could be potential risk factors for CRN, than HCV Ab (-) participants. The association between HCV and CRN disappeared after adjusting for these confounders including age. However, although no statistical significance was found in terms of an independent risk factor, patients with HCV may have a greater tendency to have CRN in terms of general association. This may reflect the suboptimal health status of patients with HCV in general. Therefore, they may still be a population at risk because there are more risk factors than non-HCV populations. Because of our results, the need for CRC screening for patients with HCV should not be overlooked. Moreover, the number of patients with HCV in our study is so small that it may be difficult to conclude from our results that HCV infection is not a risk factor for CRN. Further large-scale studies including many patients with HCV are needed to elucidate the association between HCV infection and CRN risk.

A population-based study in Taiwan revealed that patients with HBV infection exhibited an increased risk of CRC (hazard ratio (HR), 1.36; 95% CI, 1.09–1.70, *p* < 0.001), but patients with HCV infection did not (HR, 1.07; 95% CI, 0.84–1.38) [12]. Another retrospective analysis of 1413 patients with CRC demonstrated that HBV infection was an independent factor for the occurrence of liver metastasis in CRC [26]. The mechanism linking HBV infection and colorectal carcinogenesis is not well understood. The X protein from HBV is a major contributor in the progression of HBV-induced HCC and interrupts the DNA repair mechanism through the modulation of *p53* transcriptional activation [27]. Since HBV can exist in the colon [4], HBV infection may play a role in the development of CRN in a mechanism similar to that in the carcinogenesis of HBV-related HCC. The mutation of the *p53* gene is a major component in the stage of the adenoma-carcinoma sequence of CRC [28]. Thus, *p53* mutation may play an important role as a connection between the HBV infection and CRN. In addition, HBV infection has been reported to affect the modulation of the host immune response [29]. The defects in the modulation of immune responses caused by HBV may be involved in extrahepatic tumorigenesis and malignant progression, including CRC.

However, in the present study, HBV infection was not associated with the risk of advanced adenoma or CRC although it was an independent risk factor for overall CRN. The negative results for advanced adenoma or CRC might be because of the small number of patients with these lesions. Among the 5476 patients with HBsAg positivity, only 103 and 4 patients had advanced adenoma and CRC, respectively. However, our results may indicate that HBV infection might affect the early stages rather than the later stages of the adenoma-carcinoma sequence. Additional research is needed to understand the mechanisms involved in colorectal carcinogenesis in HBV-infected patients.

Our study confirmed that the well-known risk factors for CRN are also risk factors among HBV-infected patients. Age, male sex, smoking, family history of CRC, alcohol intake, and fatty liver were identified as independent risk factors for CRN in patients with HBV infection. HBV-infected patients with these risk factors may need more intensive CRC screening.

Although our study had a large sample size, there were a few limitations. First, since our study had a cross-sectional design, there was a limit to concluding a causal relationship between HBV infection and CRN. Further prospective studies are warranted to establish the causal relationship between HBV infection and the development of CRN. Second, this was not a population-based study, but rather a retrospective study that included a cohort composed of ethnic Koreans who had undergone a regular health check-up at two centers. Thus, there was likely some degree of selection bias. Our findings in healthy Korean men and women from an HBV-endemic area might not be generalizable to other populations. Finally, data on HBV viral load, antiviral treatment, disease duration, and genotype were not available, and we could not confirm whether these factors were correlated with the risk of CRN. Despite these limitations, our study adds knowledge about a topic that is currently insufficiently understood.

## 5. Conclusions

HBV infection was independently correlated with the risk of CRN. Our results suggest the possibility that HBV infection may be involved in colorectal carcinogenesis. Screening colonoscopy may have to be recommended more thoroughly for HBV-infected patients.

## Figures and Tables

**Table 1 jcm-08-02085-t001:** Baseline characteristics according to hepatitis B virus infection (total, 155,674).

Variables	HBsAg (-) (*n* = 150,198)	HBsAg (+) (*n* = 5476)	*p*
Age (years)	41.0 ± 9.1	41.6 ± 8.6	<0.001
Male sex	97,770 (65.1)	3857 (70.4)	<0.001
Current or ex-smoker ^a^	63,905 (46.4)	2428 (48.2)	0.015
Family history of CRC ^b^	6094 (4.1)	198 (3.6)	0.106
BMI (kg/m^2^) ^c^	23.7 ± 3.2	23.9 ± 3.2	<0.001
Obesity (BMI ≥30 kg/m^2^) ^c^	5280 (3.4)	213 (3.9)	0.140
Alcohol intake ≥20 g/d ^d^	35,272 (25.7)	1104 (22.3)	<0.001
Fatty liver ^e^	49,989 (33.5)	1529 (28.0)	<0.001
Hypertension	20,487 (13.6)	761 (13.9)	0.586
Diabetes mellitus	7323 (4.9)	242 (4.4)	0.123
Use of NSAID ^f^	5407 (3.6)	147 (2.7)	<0.001
Overall colorectal neoplasia	23,436 (15.6)	926 (16.9)	0.009
Any adenoma	23,382 (15.6)	922 (16.8)	0.011
Advanced adenoma	2643 (1.8)	103 (1.9)	0.503
Cancer	110 (0.07)	4 (0.07)	1.000

Data are presented as mean ± standard deviation or number (%). HBsAg, hepatitis B surface antigen; CRC, colorectal cancer; BMI, body mass index; NSAID, nonsteroidal anti-inflammatory drug. ^a^ Missing data in 13,054 participants; ^b^ Missing data in 382 participants; ^c^ Missing data in 79 participants; ^d^ Missing data in 13,622 participants; ^e^ Missing data in 910 participants; ^f^ Missing data in 245 participants.

**Table 2 jcm-08-02085-t002:** Baseline characteristics according to hepatitis C virus infection (total, 155,180).

Variables	HCV Ab (-) (*n* = 154,940)	HCV Ab (+) (*n* = 240)	*p*
Age (years)	41.0 ± 9.1	49.7 ± 10.2	<0.001
Male sex	101,192 (65.3)	126 (52.5)	<0.001
Current or ex-smoker ^a^	66,017 (46.5)	98 (47.6)	0.759
Family history of CRC ^b^	6264 (4.1)	4 (1.7)	0.065
BMI (kg/m^2^) ^c^	23.7 ± 3.2	23.8 ± 3.2	0.900
Obesity (≥30 kg/m^2^) ^c^	5465 (3.5)	9 (3.8)	0.853
Alcohol intake ≥20 g/d ^d^	36,169 (25.6)	59 (28.9)	0.274
Fatty liver ^e^	51,304 (33.3)	57 (24.2)	0.003
Hypertension	21,129 (13.6)	51 (21.3)	0.001
Diabetes mellitus	7510 (4.8)	27 (11.6)	<0.001
Use of NSAID ^f^	5526 (3.6)	10 (4.2)	0.601
Overall colorectal neoplasia	24,230 (15.6)	55 (22.9)	0.002
Any adenoma	24,173 (15.6)	54 (22.5)	0.003
Advanced adenoma	2729 (1.8)	7 (2.9)	0.208
Cancer	113 (0.07)	1 (0.42)	0.162

Data are presented as mean ± standard deviation or number (%). HCV Ab, hepatitis C antibody; CRC, colorectal cancer; BMI, body mass index; NSAID, nonsteroidal anti-inflammatory drug. ^a^ Missing data in 13,020 participants; ^b^ Missing data in 382 participants; ^c^ Missing data in 79 participants; ^d^ Missing data in 13,569 participants; ^e^ Missing data in 846 participants; ^f^ Missing data in 245 participants.

**Table 3 jcm-08-02085-t003:** Association between hepatitis B or C virus infection and risk of colorectal neoplasia.

Variables	Crude OR (95% CI)	*p*	Adjusted OR (95% CI) ^a^	*p*
Hepatitis B virus				
HBsAg (-)	1.00 (reference)		1.00 (reference)	
HBsAg (+)	1.10 (1.02–1.18)	0.009	1.10 (1.01–1.19)	0.025
Hepatitis C virus				
HCV Ab (-)	1.00 (reference)		1.00 (reference)	
HCV Ab (+)	1.30 (1.19–2.17)	0.002	1.04 (0.72–1.50)	0.839

OR, odds ratio; CI, confidence interval; HBsAg, hepatitis B surface antigen; HCV Ab, hepatitis C antibody. ^a^ Values are adjusted for age, sex, smoking status, family history of colorectal cancer, obesity (body mass index ≥ 30 kg/m^2^), alcohol intake, fatty liver, hypertension, diabetes, and use of nonsteroidal anti-inflammatory drugs.

**Table 4 jcm-08-02085-t004:** Risk factors for colorectal neoplasia in HBsAg-positive patients (*n* = 5476).

Variables	Univariable Analysis OR (95% CI)	*p*	Multivariable Analysis OR (95% CI) ^a^	*p*
Age (years)	1.07 (1.06–1.08)	<0.001	1.08 (1.07–1.09)	<0.001
Male sex	1.80 (1.51–2.14)	<0.001	1.51 (1.18–1.95)	0.001
Current or ex-smoker	2.06 (1.77–2.40)	<0.001	1.57 (1.29–1.92)	<0.001
Family history of CRC	1.75 (1.26–2.42)	0.001	1.94 (1.33–2.84)	0.001
Obesity (BMI ≥ 30 kg/m^2^)	1.11 (0.78–1.58)	0.581	0.93 (0.62–1.39)	0.719
Alcohol intake ≥ 20 g/d	1.66 (1.41–1.96)	<0.001	1.23 (1.02–1.49)	0.028
Fatty liver	1.45 (1.26–1.68)	<0.001	1.24 (1.04–1.48)	0.019
Hypertension	1.72 (1.43–2.07)	<0.001	0.96 (0.77–1.21)	0.738
Diabetes mellitus	2.04 (1.53–2.72)	<0.001	1.16 (0.82–1.63)	0.402
Use of NSAID	1.11 (0.73–1.69)	0.635	1.08 (0.66–1.79)	0.753

OR, odds ratio; CI, confidence interval; CRC, colorectal cancer; BMI, body mass index; NSAID, nonsteroidal anti-inflammatory drug; ^a^ Values are adjusted for age, sex smoking status, family history of colorectal cancer, obesity, alcohol intake, fatty liver, hypertension, diabetes, and use of nonsteroidal anti-inflammatory drugs.

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
