# Peer review of "Correlation between Hepatitis B Virus Infection and Colorectal Neoplasia"

_jcm, 2019, doi:10.3390/jcm8122085_

Round 1

Reviewer 1 Report

1) change the title. Current title is not attractive

2) Exclude results on HCV population from paper. The population is too limited for strong conclusions.

3) Please add impact of HBV viral load and HBV treatment on the prevalence of adenoma.These data will add novelty and originality and will promote the interest to the reader

4) line 159: add statistical data in the text

5) Prefer "association" to "correlation" when describe a possible link between HBV and adenoma

6) line 263-264: add p value

7) change the discussion section on the basis of point 2) and 3) 

8) table 1. Obesity should be BMI≥30 Kg/m2

Author Response

Reviewer 1

1) change the title. Current title is not attractive

Reply: Thank you for your comment. We modified the title as follows: “Correlation between Hepatitis B virus infection and colorectal neoplasia”

2) Exclude results on HCV population from paper. The population is too limited for strong conclusions.

Reply: Thank you for this comment, we agree with your concern. However, as shown below, the second reviewer emphasized the clinical implications of the results on HCV population. Therefore, we could not delete the results for HCV. Instead, we deleted the results for HCV from the title and the conclusions of abstract and manuscript. In addition, we weakened the results for HCV in the discussion section as follows: “Moreover, the number of patients with HCV in our study is so small that it may be difficult to conclude from our results that HCV infection is not a risk factor for CRN. Further large-scale studies including many patients with HCV are needed to elucidate the association between HCV infection and CRN risk.”

3) Please add impact of HBV viral load and HBV treatment on the prevalence of adenoma. These data will add novelty and originality and will promote the interest to the reader

Reply: Unfortunately, data on HBV viral load and HBV treatment were not available, and thus we could not evaluate the impact of viral load and antiviral treatment on the prevalence of CRN. We added this limitation in the Discussion section.

4) line 159: add statistical data in the text

Reply: Per the reviewer’s recommendation, we have added statistical data as follows: “However, the prevalences of advanced adenoma and CRC were not different between HBsAg (-) and HBsAg (+) participants (advanced adenoma: 1.8% vs. 1.9%, P = 0. 503 and CRC: 0.07% vs. 0.07%, P = 1.000).”

5) Prefer "association" to "correlation" when describe a possible link between HBV and adenoma

Reply: We replaced “association" and “associated” with “correlation" and “correlated” when describe a possible link between HBV and adenoma in our results.

6) line 263-264: add p value

Reply: As the reviewer recommended, we have added p value as follows: “A population-based study in Taiwan revealed that patients with HBV infection exhibited an increased risk of CRC (hazard ratio [HR], 1.36; 95% CI, 1.09–1.70, P < 0.001), but patients with HCV infection did not (HR, 1.07; 95% CI, 0.84–1.38).”

However, this Taiwan paper did not present p value for “HCV infection (HR, 1.07; 95% CI, 0.84–1.38)”

7) change the discussion section on the basis of point 2) and 3) 

Reply: As mentioned above, we weakened the results for HCV in the discussion section and deleted the results for HCV from the conclusion of the discussion section. Additionally, we mentioned the following limitation in the discussion section: “data on HBV viral load, antiviral treatment, disease duration, and genotype were not available, and we could not confirm whether these factors were correlated with the risk of CRN.”

8) table 1. Obesity should be BMI≥30 Kg/m2

Reply: As the reviewer recommended, we changed “Obesity (≥25 kg/m2)” to “Obesity (≥30 kg/m2)” in Tables 1, 2, 3, and 4.

We appreciate the time that you have taken to review our manuscript.

Reviewer 2

This is a very interesting and well written paper.  The number of patients with hepatitis B is large yet the number of patients with hepatitis C is very small (n= 250).  This is in contrast with North America where the number of patients with HCV is high yet the number of patients with HBV is small and usually confined to specific communities (e.g. the Asian community).  Hence, the generalizabilty of this study outside of Asia is potentially challenging.  The analysis is very well done and I do not have any specific comments.  The major comment is in regards to the finding of an increased incidence of colorectal adenoma in the HCV group vs non-HCV group that was statistically significant in the initial analysis but after controlling for confounders (all of which are probably risk factors for colorectal neoplasia) was not significant.  The investigators conclude that HCV is not a risk factor for colorectal neoplasia.  Although in terms of an independent risk factor (that may suggest a hypothesis of a causal association) statistical significance was not found one can argue that in terms of a general association, yes, the HCV patient population may have a greater tendency to have colorectal neoplasia and that may reflect the general suboptimal health of HCV patients in general. Therefore, this may still be a population at risk if only because they have more risk factors than the non-HCV population.  One can speculate that HCV may still be a reason to screen with non-invasive tests (ie. fecal occult blood test with the FIT test) if clinicians are going to be selective about who they screen.

Reply: We appreciate the reviewers’ insightful comments. To reflect the reviewer's comments, we added the following sentences in the Discussion section: “In our study, the prevalence of CRN was higher in HCV Ab (+) participants than in HCV Ab (-) participants (22.9% vs. 15.6%, P = 0.002). However, these results could be because HCV Ab (+) participants were much older than HCV Ab (-) participants (mean age, 50 vs. 41 years). Age is a potent risk factor for CRN [25]. Additionally, HCV Ab (+) participants had a higher prevalence of diabetes and hypertension, which could be potential risk factors for CRN, than HCV Ab (-) participants. The association between HCV and CRN disappeared after adjusting for these confounders including age. However, although no statistical significance was found in terms of an independent risk factor, patients with HCV may have a greater tendency to have CRN in terms of general association. This may reflect the suboptimal health status of patients with HCV in general. Therefore, they may still be a population at risk because there are more risk factors than non-HCV populations. Because of our results, the need for CRC screening for patients with HCV should not be overlooked. Moreover, the number of patients with HCV in our study is so small that it may be difficult to conclude from our results that HCV infection is not a risk factor for CRN. Further large-scale studies including many patients with HCV are needed to elucidate the association between HCV infection and CRN risk.”

We appreciate the time that you have taken to review our manuscript.

Reviewer 2 Report

This is a very interesting and well written paper.  The number of patients with hepatitis B is large yet the number of patients with hepatitis C is very small (n= 250).  This is in contrast with North America where the number of patients with HCV is high yet the number of patients with HBV is small and usually confined to specific communities (e.g. the Asian community).  Hence, the generalizabilty of this study outside of Asia is potentially challenging.  The analysis is very well done and I do not have any specific comments.  The major comment is in regards to the finding of an increased incidence of colorectal adenomata in the HCV group vs non-HCV group that was statistically significant in the initial analysis but after controlling for confounders (all of which are probably risk factors for colorectal neoplasia) was not significant.  The investigators conclude that HCV is not a risk factor for colorectal neoplasia.  Although in terms of an independent risk factor (that may suggest a hypothesis of a causal association) statistical significance was not found one can argue that in terms of a general association, yes, the HCV patient population may have a greater tendency to have colorectal neoplasia and that may reflect the general suboptimal health of HCV patients in general. Therefore, this may still be a population at risk if only because they have more risk factors than the non-HCV population.  One can speculate that HCV may still be a reason to screen with non-invasive tests (ie. fecal occult blood test with the FIT test) if clinicians are going to be selective about who they screen.

Author Response

Reviewer 2

This is a very interesting and well written paper.  The number of patients with hepatitis B is large yet the number of patients with hepatitis C is very small (n= 250).  This is in contrast with North America where the number of patients with HCV is high yet the number of patients with HBV is small and usually confined to specific communities (e.g. the Asian community).  Hence, the generalizabilty of this study outside of Asia is potentially challenging.  The analysis is very well done and I do not have any specific comments.  The major comment is in regards to the finding of an increased incidence of colorectal adenoma in the HCV group vs non-HCV group that was statistically significant in the initial analysis but after controlling for confounders (all of which are probably risk factors for colorectal neoplasia) was not significant.  The investigators conclude that HCV is not a risk factor for colorectal neoplasia.  Although in terms of an independent risk factor (that may suggest a hypothesis of a causal association) statistical significance was not found one can argue that in terms of a general association, yes, the HCV patient population may have a greater tendency to have colorectal neoplasia and that may reflect the general suboptimal health of HCV patients in general. Therefore, this may still be a population at risk if only because they have more risk factors than the non-HCV population.  One can speculate that HCV may still be a reason to screen with non-invasive tests (ie. fecal occult blood test with the FIT test) if clinicians are going to be selective about who they screen.

Reply: We appreciate the reviewers’ insightful comments. To reflect the reviewer's comments, we added the following sentences in the Discussion section: “In our study, the prevalence of CRN was higher in HCV Ab (+) participants than in HCV Ab (-) participants (22.9% vs. 15.6%, P = 0.002). However, these results could be because HCV Ab (+) participants were much older than HCV Ab (-) participants (mean age, 50 vs. 41 years). Age is a potent risk factor for CRN [25]. Additionally, HCV Ab (+) participants had a higher prevalence of diabetes and hypertension, which could be potential risk factors for CRN, than HCV Ab (-) participants. The association between HCV and CRN disappeared after adjusting for these confounders including age. However, although no statistical significance was found in terms of an independent risk factor, patients with HCV may have a greater tendency to have CRN in terms of general association. This may reflect the suboptimal health status of patients with HCV in general. Therefore, they may still be a population at risk because there are more risk factors than non-HCV populations. Because of our results, the need for CRC screening for patients with HCV should not be overlooked. Moreover, the number of patients with HCV in our study is so small that it may be difficult to conclude from our results that HCV infection is not a risk factor for CRN. Further large-scale studies including many patients with HCV are needed to elucidate the association between HCV infection and CRN risk.”

We appreciate the time that you have taken to review our manuscript.
